# Oxidative Stress and DNA Damage Biomarkers in Heart Failure: A Systematic Review and Meta-Analysis

**DOI:** 10.3390/antiox14101249

**Published:** 2025-10-17

**Authors:** Francesca Milani, Annamaria Porreca, Giuseppe Rosano, Laura Vitiello, Maurizio Volterrani, Patrizia Russo, Stefano Bonassi

**Affiliations:** 1Department of Human Sciences and Quality of Life Promotion, San Raffaele University, Via di Val Cannuta 247, 00166 Rome, Italy; francesca.milani@uniroma5.it (F.M.); annamaria.porreca@uniroma5.it (A.P.); giuseppe.rosano@gmail.com (G.R.); laura.vitiello@uniroma5.it (L.V.); maurizio.volterrani@sanraffaele.it (M.V.); stefano.bonassi@uniroma5.it (S.B.); 2Clinical and Molecular Epidemiology, Istituto di Ricovero e Cura a Carattere Scientifico (IRCCS) San Raffaele Roma, Via di Val Cannuta 247, 00166 Rome, Italy; 3Clinical and Experimental Research Center, Istituto di Ricovero e Cura a Carattere Scientifico (IRCCS) San Raffaele Roma, 00166 Rome, Italy; 4Cardiology Rehabilitation Unit, Istituto di Ricovero e Cura a Carattere Scientifico (IRCCS) San Raffaele Roma, Via della Pisana 235, 00163 Rome, Italy

**Keywords:** heart failure, ejection fraction, oxidative stress, DNA damage, meta-analysis

## Abstract

Background: Oxidative stress is a key driver of heart failure (HF) pathophysiology, promoting myocardial injury, inflammation, and remodeling. Although numerous biomarkers of oxidative stress and DNA damage have been investigated in HF, their clinical relevance remains uncertain. This systematic review and meta-analysis aimed to evaluate alterations in these biomarkers in HF patients compared to healthy controls. Methods: A comprehensive search of PubMed, MEDLINE, the Cochrane Library, and Web of Science was conducted in accordance with PRISMA guidelines. Studies reporting oxidative stress or DNA damage biomarkers in HF patients versus controls were included. Random-effects models were used to calculate ratios of means (ROM) with 95% confidence intervals (CI). Heterogeneity and publication bias were assessed using the I^2^ statistic and Begg’s test. Results: Data from 3015 HF patients and 2704 controls were analyzed. HF patients had significantly higher levels of 8-hydroxy-2′-deoxyguanosine (8-OHdG) (ROM = 2.24, 95% CI: 1.75–2.88), malondialdehyde (MDA) (ROM = 1.87, 95% CI: 1.49–2.36) and isoprostanes (ROM = 2.83, 95% CI: 1.97–4.05). Telomere length was significantly shorter (ROM = 0.66, 95% CI: 0.53–0.81), indicating accelerated cellular aging. Considerable heterogeneity was observed across studies. Conclusion: This meta-analysis supports a robust association between oxidative stress, DNA damage, and HF, highlighting the potential role of these biomarkers in disease monitoring and prognosis.

## 1. Introduction

Heart failure (HF) is a clinical syndrome where the heart is unable to either pump blood effectively or adequately fills with blood. According to a most recent consensus statement, symptoms and signs of HF are caused by a structural and/or functional cardiac abnormality and corroborated by elevated natriuretic peptide levels and or objective evidence of pulmonary or systemic congestion [1]. Heart failure is traditionally classified in three classes according to the ejection fraction (EF), i.e., preserved ejection fraction (HFpEF), reduced ejection fraction (HFrEF), and mid-range or mildly reduced ejection fraction (HFmrEF).

HFpEF is characterized by an ejection fraction (EF) greater than 50% and is associated with various pro-inflammatory and metabolic comorbidities. It is characterized by structural and cellular changes such as cardiomyocyte hypertrophy, fibrosis, and inflammation, resulting in impaired relaxation of the left ventricle. In contrast, HFrEF is defined by an EF less than 40% and is marked by significant cardiomyocyte loss leading to systolic dysfunction, where the left ventricle struggles to contract effectively. HFmrEF represents an intermediary stage with an EF between 40 and 49%, often progressing to either HFpEF (25% of cases) or HFrEF (33% of cases) [2]. Data from population-based registries indicate that roughly half of HF patients present with HFrEF [3].

The pathophysiology of HF is characterized by a ventricular dysfunction that leads to reduced cardiac output, activating compensatory neurohormonal mechanisms such as the renin–angiotensin–aldosterone system and the sympathetic nervous system. These mechanisms, in the long term, lead to unfavorable ventricular remodeling and further damage to the myocardium. Additionally, there is an alteration of endothelial function, with consequent oxidative stress and inflammation, contributing to the progression of the disease [4].

Oxidative stress, mediated by reactive oxygen species (ROS) and reactive nitrogen species (RNS), is a significant factor in the development of cardiac fibrosis, a key element in the advancement of HF [5]. This process is characterized by an imbalance between the generation of ROS and the body’s natural antioxidant defense systems, known as the ‘redox state’. While moderate amounts of ROS play a vital role in maintaining cellular equilibrium, an overabundance can result in cellular malfunction, proteins and lipids oxidation, genetic damage, and ultimately irreversible cellular injury and demise [6].

The generation of ROS within cardiac tissue is primarily facilitated by mitochondria, NADPH oxidases, xanthine oxidase, and uncoupled nitric oxide synthase (NOS). Under pathological conditions, the mitochondrial electron transport chain triggers the production of substantial amounts of superoxide, contributing to damage in cardiomyocytes and exacerbating myocardial injury following acute myocardial infarction [7]. Furthermore, the production of ROS is heightened by the upregulation of NADPH oxidase expression and activity in response to pathological triggers such as mechanical strain, angiotensin II, endothelin-1, and tumor necrosis factor-alpha (TNF-α). Similarly, the expression and activity of xanthine oxidase are elevated in the failing heart, leading to increased ROS production. Cardiac injury also results in the uncoupling and structural instability of NOS, promoting heightened ROS generation [8].

Moreover, HF is marked by a significant escalation in oxidative stress generation, alongside a depletion of the intrinsic antioxidant defense system. In cardiomyocytes, as in most cell types, the primary endogenous constituents of the antioxidant defense system responsible for neutralizing ROS include superoxide dismutase (SOD), catalase, glutathione peroxidase (GPx), nicotinamide adenine dinucleotide (NAD+), and glutathione (GSH) [9].

To evaluate oxidative stress and the consequent DNA damage, a range of biomarkers present in blood and urine can be utilized. These markers indicate the extent of oxidative stress within the organism. Biomarkers are measurable characteristics that serve as indicators of health, disease, or the body’s response to exposure or therapeutic interventions.

In the context of cardiovascular disease (CVD), the last 30 years have witnessed significant progress in biomarker research. This has led to the development of highly sensitive screening tools, enhanced emphasis on early diagnosis, and more effective therapeutic approaches. These advancements have collectively contributed to better clinical outcomes and improved patient care [10]. A 2019 FDA guidance document indicates that biomarkers have potential utility to enroll heart failure (HF) patients with a greater event risk, stratify patients based on their predicted prognosis, and allow early proof of concept and dose selection studies [10]. Despite the almost exclusive focus on B-type natriuretic peptide (BNP) and N-terminal proBNP (NT-proBNP), many biomarkers reflecting different elements of HF pathophysiology exist [11], and the identification of specific biomarkers for each drug has been proposed [12].

A growing body of evidence supports the association between viral infections and cardiovascular diseases (CVD). Cytomegalovirus (CMV) infection, for instance, has been implicated in the pathogenesis of cardiovascular disorders [13].

Likewise, SARS-CoV-2 infection has been reported to increase the risk of CVD, with clinical manifestations including acute myocardial injury or infarction, myocarditis, heart failure, and arrhythmias [14].

Notably, elevated levels of Torque Teno Virus (TTV) have been correlated with a higher risk of ischemic heart disease (IHD) in the elderly. A study involving 900 non-IHD individuals and 86 patients with IHD, aged 55–75 years, demonstrated that increased TTV burden was associated with IHD prevalence [15].

TTV is a small, non-enveloped, single-stranded DNA virus of the *Anelloviridae* family, highly prevalent in the human population and representing a dominant constituent of the human virome [16].

Our previous investigations revealed that TTV viremia ≥ 4log10 copies/mL was linked to elevated DNA damage (as determined by comet assay), increased mortality, and immune dysfunction in patients with chronic obstructive pulmonary disease (COPD) [17].

Preliminary findings from our ongoing studies further support the hypothesis that TTV viremia above this threshold may also be associated with cardiovascular pathologies [Russo et al., ongoing study] [18].

Various biomarkers have been identified and utilized to evaluate these processes, for example, malondialdehyde (MDA) is a highly reactive dialdehyde generated as an end-product of lipid peroxidation of membrane polyunsaturated fatty acids (PUFAs) in response to free radicals and ROS [19]. Additionally, 8-hydroxy-2′-deoxyguanosine (8-OHdG) is a predominant form of ROS lesions found in both nuclear and mitochondrial DNA, often used as a biomarker for oxidative stress [20]. Another commonly used biomarker is myeloperoxidase (MPO). Human MPO is a homodimeric protein that contributes to oxidative stress through the production of ROS and other oxidative molecules such as HOCl [21]. Several biomarkers are used in research to quantify and identify oxidative stress and DNA damage, as well as inflammation and other pathological processes. In the field of HF, these biomarkers have been discovered and used over the past 30 years, allowing for a deeper understanding of the molecular and cellular dynamics that contribute to the progression of the disease. The present systematic review and meta-analysis sought to critically evaluate published evidence on biomarkers related to oxidative stress in HF patients. Through the analysis of these biomarkers, it is possible to reconstruct the pathophysiology of oxidative stress and DNA damage in the context of HF and understand how these processes contribute to the exacerbation of the disease. This systematic review and meta-analysis aim to critically evaluate the current evidence on biomarkers of oxidative stress and DNA damage in HF patients. By quantifying alterations in these biomarkers, the study seeks to elucidate their pathophysiological roles in HF progression, distinguish their relevance across different HF phenotypes, and assess their potential utility for diagnosis, prognosis, and therapeutic monitoring. Through this analysis, it is also possible to reconstruct the mechanisms underlying oxidative stress and DNA damage in HF and to understand how these processes contribute to disease exacerbation. This study is innovative as it provides the first systematic meta-analysis of oxidative stress and DNA damage biomarkers in heart failure, offering a multidimensional view beyond natriuretic peptides. By integrating genomic, lipid, and protein markers, it advances understanding of disease mechanisms and prognostic potential. The inclusion of genomic instability markers highlights new opportunities for risk stratification, personalized treatment, and precision rehabilitation.

## 2. Materials and Methods

Studies included in this systematic review and meta-analysis were sourced from PubMed, MEDLINE, the Cochrane Library, and Web of Science databases, according to the PRISMA (Preferred Reporting Items for Systematic Reviews and Meta-Analyses) guidelines. The research was conducted separately for each biomarker associated with DNA damage and oxidative stress. Specific search terms were used to ensure comprehensive coverage of relevant studies. For the disease context, the following terms were applied: ‘heart failure’, ‘HF’, ‘cardiac failure’, ‘cardiomyopathy’, ‘congestive heart failure’, ‘left ventricular dysfunction’, ‘CHF’, ‘chronic heart failure’, ‘dilated cardiomyopathy’, ‘chronic heart disease’, ‘ischemic heart disease’, and ‘myocardial infarction’. For biomarkers, the search terms were 8-hydroxy-2′-deoxyguanosine ‘8-oxo-guanine’, ‘8-oxo-dg’, ‘8-hydroxy-2-deoxyguanosine’, ‘8ohdg’, ‘8-OH-dg’, ‘8-OHG’, ‘8-oxo-G’, ‘8-hydroxydeoxyguanosine’, ‘8-hydroxyguanine’, ‘8-oxo-2-deoxyguanosine’, and ‘8-oxo-7,8-dihydro-2-deoxyguanosine’; telomeres ‘telomeres’, ‘telomerase’, and ‘telomere length’; malondialdehyde ‘malondialdehyde’ and ‘MDA’; isoprostanes: ‘isoprostanes’, ‘8-iso-PGF2α’, ‘8-isoprostaglandin F2α’, and ‘F2-isoprostanes’; nitrotyrosine: ‘nitrotyrosine’, ‘3-nitrotyrosine’, ‘3-NT’, and ‘3nt’; myeloperoxidase: ‘myeloperoxidase’ and ‘MPO’; comet assay: ‘comet assay’, ‘single-cell gel electrophoresis’, and ‘DNA damage assay’, micronuclei: ‘micronuclei’, ‘micronucleus assay’, and ‘cytokinesis-block micronucleus assay’. This systematic review and meta-analysis were conducted in accordance with the PRISMA guidelines, ensuring transparency in study identification, screening, eligibility and inclusion.

### 2.1. Eligibility Criteria

Eligible studies were original human research articles comparing biomarkers of oxidative stress or DNA damage between heart failure patients and healthy controls. Inclusion criteria required: (i) adult population (≥18 years); (ii) a clearly defined HF diagnosis (HFrEF, HFpEF, or HFmrEF); (iii) quantitative assessment of biomarkers with reported means ± SD or medians with IQR. Most of the eligible articles were cross-sectional in nature, comparing biomarker levels at a single time point between HF patients and healthy controls. Exclusion criteria were: (i) animal or in vitro studies; (ii) non-original articles (e.g., reviews, editorials, commentaries); (iii) studies lacking a control group or reporting biomarkers unrelated to oxidative stress or DNA damage; (iv) non-English publications

### 2.2. Study Outcome

The primary objective of this study was to evaluate the presence of oxidative stress and DNA damage in HF patients as compared to healthy controls. To achieve this objective, we focused on biomarkers linked to oxidative stress and DNA damage most commonly investigated in the literature, i.e., 8-hydroxy-2′-deoxyguanosine (8-OHdG), malondialdehyde (MDA), telomere length, isoprostanes, nitrotyrosine, myeloperoxidase (MPO), DNA damage (Comet assay), and micronuclei.

Based on the collected data, several meta-analyses were conducted to assess the heterogeneity of outcomes. The analysis compared the levels of biomarkers between patients and a healthy control group. As regards the Comet assay and micronuclei, only one study was identified; therefore, these biomarkers will be addressed exclusively within the systematic review and not considered for the meta-analysis. The references of all retrieved articles were examined to identify other eligible studies not indexed in the mentioned databases. Furthermore, an examination of the articles identified was conducted using Artificial Intelligence, specifically through the website https://www.researchrabbit.ai/ (accessed on 23 October 2024), to identify any additional studies that may have been overlooked. No additional relevant articles were found. In cases where biomarkers were reported using different measurement units across studies (e.g., ng/mL, µmol/L, or per creatinine in urine), conversions were performed based on available information within the studies themselves or according to the most frequently used unit in the majority of included studies.

### 2.3. Statistical Analysis

Descriptive statistics of single studies were presented as mean and standard deviation. For studies reporting only median, the method proposed by Wan et al. (2014) [22] was used to transform medians into means and estimate standard deviations. All statistical analyses were performed using the R environment (version 4.0.3), using the metacont function from the meta package.

Given that the assumptions for using a fixed-effect model are often too restrictive, a random-effects model generally was applied. This approach accounts for between-study variability arising from differences in treatment, population, study design, and data analysis methods. The measure of effect estimated by random-effects models to compare biomarker levels between the two groups with the Ratio of Means (ROM). Heterogeneity was quantified using the I^2^ statistic, accompanied by the τ^2^ statistic to estimate between-study variance. Based on the χ^2^ statistic, the Q test was used to detect heterogeneity, applying a significance level of 0.10 due to the reduced power of this test in meta-analyses with a limited number of studies.

The inverse-variance method was adopted for pooling results, ensuring robust estimates for continuous outcomes. Funnel plots were generated using the funnel function from the metafor package to assess potential publication bias. Begg’s test was used to evaluate asymmetry in funnel plots, although this test has limited robustness in meta-analyses with a small number of studies. A significance level of 0.05 was applied. Sensitivity analysis was conducted to assess the robustness of the effect estimates. Individual studies were sequentially excluded, and the pooled estimates were recalculated, to identify influential studies or those contributing disproportionately to overall heterogeneity. Subgroup analyses by heart failure phenotype (HFrEF vs. HFpEF) were performed only for the MPO, as this was the only marker with a sufficient number of studies reporting data separately for both subtypes. For all other biomarkers, the available studies either focused exclusively on patients with HFrEF or did not provide stratified results by HF phenotype, making subgroup comparisons unfeasible.

## 3. Results

### 3.1. Data Collection

A detailed description of the study selection process for the biomarkers included in the meta-analysis is provided in the flowchart (Figure 1).

A total of 1664 articles were identified through database searching (up to October 2024).

After categorization by biomarker 135 studies investigated 8-OHdG, 455 MDA, 177 telomeres, 50 isoprostanes, 366 MPO and 554 nitrotyrosine.

Following screening, reviews, meta-analyses, and animal studies were excluded (73 for 8-OHdG, 280 for MDA, 57 for telomeres, 15 for isoprostanes, 156 for MPO, and 413 for nitrotyrosine). Additional exclusions were made for studies without a control group or focused on diseases other than HF (45 for 8-OHdG, 154 for MDA, 109 for telomeres, 25 for isoprostanes, 181 for MPO, and 129 for nitrotyrosine).

A further 49 studies were excluded at the eligibility stage because endpoints did not match the research question (10 for 8-OHdG, 8 for MDA, 7 for telomeres, 1 for isoprostanes, 16 for MPO, and 7 for nitrotyrosine). In addition, 7 studies were excluded due to the absence of mean ± SD or median and IQR values (2 for MDA, 2 for isoprostanes and 4 for MPO).

Finally, 44 studies were included in the quantitative synthesis: 7 on 8-OHdG, 12 on MDA, 4 on telomeres, 7 on isoprostanes, 9 on MPO, and 5 on nitrotyrosine (Figure 1). These studies contributed a total of 3015 HF patients and 2704 controls to the meta-analysis. When studies reported multiple biomarkers, each biomarker was analyzed separately, but individual participants were counted only once in the overall totals.

Table 1 reports the exact number of HF patients and controls included in the analysis for each biomarker.

### 3.2. Biomarkers of Oxidative Stress and DNA Damage in HF Patients

The meta-analysis could not be performed for some biomarkers, due to limited available data. Nevertheless, they remain crucial as they laid the foundation for our review and underscore the potential for further research in this area. DNA damage was assessed through the Comet Assay in 40 patients with HF due to dilated cardiomyopathy (DCM) and compared to 40 healthy controls.

The results revealed that HF patients with DCM exhibited significantly elevated DNA damage compared to the controls. The mean tail intensity for the HF group was 23.5 ± 7.3, more than twice the corresponding value in the healthy controls, i.e., 10.2 ± 3.4 (*p* < 0.001) [23].

Tubić Vukajlović and colleagues [24], analyzed DNA damage using the Comet Assay in the peripheral blood lymphocytes of a cohort of 50 HF patients vs. 50 healthy controls. The level of DNA damage in HF patients, i.e., 19.6 ± 6.2 was significantly higher than in that measured in healthy controls, i.e., 8.4 ± 2.1 (*p* < 0.001) [24]. This study evaluated also chromosomal instability by analyzing micronuclei frequency—another indicator of DNA damage—in the same groups. The frequency of micronuclei was significantly higher in the HF group, with a mean of 5.3 ± 1.2 per 1000 cells, compared to 1.2 ± 0.4 per 1000 cells in the control group (*p* < 0.001).

Another biomarker used to measure DNA damage is the γ-H2AX, a protein that plays a crucial role in the DNA damage response, particularly in double-strand breaks (DSBs). During the DNA repair process, H2AX acts as a key factor that accumulates at the site of damage. After phosphorylation, γ-H2AX can be detected through various techniques, making it a valuable marker for assessing DNA damage. Mondal and colleagues studied 30 HF patients admitted for receiving a continuous flow left ventricular assist device (LVAD), comparing them with 15 healthy controls. Before LVAD implantation, the HF patients showed 5.81 ± 2.19 γ-H2AX foci per nucleus, whereas the healthy controls had 2.32 ± 1.02 foci per nucleus (*p* < 0.001). The authors suggest that HF is associated with increased DNA damage, further emphasizing the potential role of—oxidative stress in the pathophysiology of heart failure [25].

#### 3.2.1. Meta-Analyses of Biomarkers of Oxidative Stress in HF Patients

In total, the meta-analysis included a total of 3015 HF patients and 2704 controls, with a male predominance (~65–70%). The mean age of the patients was 62.5 ± 12.5 years (range 46–82 years). Most patients had heart failure with reduced ejection fraction, while a smaller proportion had heart failure with preserved ejection fraction, with the majority classified as NYHA III-IV.

The etiology of HF was predominantly ischemic heart disease (~40–50%) and dilated cardiomyopathy (both DCM/IDCM) (~30–40%), with a smaller percentage of patients affected by hypertensive heart disease (~10–15%), valvular heart disease (~5–10%), and very rare congenital heart disease.

The most frequently used pharmacological treatments included ACE inhibitors/ARBs (82%), beta-blockers (70%), diuretics (80–90%), aldosterone antagonists (40–55%), digoxin (40–50%), anticoagulants (30–40%), antiplatelet agents (40%), statins (50–65%) and nitrates (10–20%). A comprehensive description of the study groups from papers included in the meta-analysis is provided in Table 2 [25,26,27,28,29,30,31,32,33,34,35,36,37,38,39,40,41,42,43,44,45,46,47,48,49,50,51,52,53,54,55,56,57,58,59,60,61].

**Table 2 antioxidants-14-01249-t002:** Demographic and Clinical Characteristics of Heart Failure Patients included in the Meta-Analysis.

Study, Year	Population, Sex (M/F)	Mean, Age, Years (SD/IC)	HF Classification	HF Aethiology	Pharmacological Treatment	Sample	Biomarker	Results	*p* Value
**8-OHdG**									
[26]	HF 78 (57 M/21 F) Controls 12	HF 64 ± 14	HFrEF	IHD, DCM, HCM	Diuretics 74%; ACEi 73%; BB 51%; anti-aldosterone agents 42%; digoxin 26%; CAs 10% and ARBs 16%	8-OHdG	Blood sample	HF: 0.34 ± 0.54 Controls: 0.04 ± 0.07	<0.05
[27]	HF 194 (108 M/86 F) Controls 31 (20 M/11 F)	HF 57.1 ± 14.4 Controls 52.5 + 13.2	HFrEF	DCM, MI	BB 80%, ACE-I 70%, 23% ARB, LDs 78%, 57% aldosterone antagonist, e 21% statin.	8-OHdG	Urine sample	HF: 13.0 ± 5.7 Controls: 8.2 ± 1.9	<0.0001
[28]	HF 230 (140 M/90 F) Controls 42 (27 M/15 F)	HF 70.3 ± 12.8	HFrEF and HFpEF	DCM, IHD, HVD, HHD, HVD	73% ACEi; 51% BB; CCBs 19%; 67% Diuretics; Statins 16%	8-OHdG	Blood sample	HF: 0.40 ± 0.24 Controls: 0.22 ± 0.09 ng/mL	<0.001
[29]	HF 56 (42 M/14 F) Controls 20	HF 53 ± 12	HFrEF and HFpEF	DCM	71% ACEi; ARBs 22%; CCBs 23%; Diuretics 55%	8-OHdG	Blood sample	HF: 5.2 ± 2.9; Controls: 3.0 ± 1.5	0.0018
[30]	HF 111 (M 52%/F 48%) Controls 30	HF 57 ± 16	HFrEF and HFpEF			8-OHdG	Urine sample	HF: 14.6 ± 9.2, Controls: 6.8 ± 1.9	<0.01
[31]	25 HF and 33 Controls	HF Mean 69	HFrEF	CHD		8-OHdG	Urine sample	HF: 33.7 ± 4.0 Controls: 12.6 ± 0.9	<0.01
[32]	HF (24 M/8 F) Controls (8 M/6 F)	HF 46.6 ± 18.2 Controls 34.6 ± 6.9	HFrEF	DCM	84% Diuretics; ACEi/ARBs 94%; BB 78%; Digoxin 38%	8-OHdG	Blood sample	HF: 0.75 ± 0.57 Controls: 0.23 ± 0.07	0.003
**T** **elomere**									
[33]	HF 620 (493 M e 127 F) 183 Controls (145 M e 38 F)	HF 66.2 ± 8.9 Controls 66.2 ± 8.7	HFrEF	ICHF and NICHF	51% BB; 88% ACEi; 10% ARBs; 93% Diuretics	Telomere	Blood samples	HF: 0.64 ± 0.30 Controls: 1.05 ± 0.32	<0.001
[34]	HF 34 (24 M/10 F) Controls 46 (31 M/15 F)	HF 56 ± 9 Controls 46 ± 16	HFrEF	IDCM		Telomere	Blood samples	IDCM: 6.97 ± 0.63 Controls: 9.11 ± 2.12	<0.001
[35]	CHF 27 (23 M/4 F) Controls 24 (20 M/4 F)	HF 66 ± 6.6 Controls 69 ± 6.9	HFrEF		100% ACEI/ARB; 96% BB; 81% Diuretics; 88% Statins; 96% anticoagulants; 25% aldosterone antagonists	Telomere	Blood samples	HF: 0.55 ± 0.074 Controls: 0.70 ± 0.074	0.002
[36]	40 CHF (17 M e 23 F) Controls 40 (18 M e 22 F)	HF 82 (77–89) Controls 80 (76–85)	HFrEF	ICHF		Telomere	leukocytes	HF: 0.18 ± 0.08 Controls: 0.38 ±0.25	<0.01
**Malondialdehyde**									
[37]	HF 58 (43 M/15 F) Controls 17 (12 M/7 F)		HFrEF	IDCM, IHD	76% ACEI; 24% BB; 79% Diuretics; 58% Digoxin; 41% ASA; 15% Amiodarone	MDA	Blood sample	HF: 0.65 ± 0.10 Controls: 0.25 ± 0.05	<0.005
[38]	HF 12 (10 M/2 F) Controls 6 (6 M)	HF (mean 52) Controls (mean 23)	HFrEF and HFpEF	HVD, HHD, DCM	100% diuretics; 100% digitalis	MDA	Blood sample	HF: 3.7 ± 1.3 Controls: 1.9 ± 0.6	<0.01
[39]	HF 12 (M10/F2) Controls 4 (4 M)	DCM (54 ± 2) Controls (18, 25–27, 37)	HFrEF	DCM		MDA	Heart tissue	HF: 152 ± 38 Controls: 74 ± 6	
[40]	HF 109 (93 M/16 F) Controls 28	HF (45.97 ± 10.8)	HFrEF	DCM	94.5% BB, 92.66% ACEIs, 35.78% ARBs, 90.83% MRAs, 13.76% Amiodarone, LDs 60.55%, TDs 18.35%, OACs 44.95%, 63.30% digoxin	MDA	Blood sample	HF: 4.37 (3.68–5.78) Controls: 1.31 (1.14–1.41)	<0.001
[41]	HF 10 (7 M/3 F) Controls 69 (40 M/29 F)	61.6 ± 5.5	HFrEF		81.8% BB, 36.4% CCBs, 81.8% aspirin, 63.9% nitrates, 90.9% ACEi, 9.1% ARBs, 54.5% statins, 90.9% spironolactone, furosemide	MDA	Blood sample	HF: 8.5 ± 0.6 Controls: 7.3 ± 0.9	<0.05
[42]	HF 30 (30 M) Controls 16 (14 M/2 F)	HF 63 ± 8.2 Controls 53.8 ± 15.7	HFrEF		76.7% ACEI, 87.7% nitrates, 63.3% diuretics, 73.3% digitalis, 3% hydralazine	MDA	Blood sample	HF: 2.65 ± 1.3 Controls: 1.45 ± 0.77	<0.05
[43]	CHF 45 (37 M/8 F) Controls 45 (29 M/16 F)	HF 58 range (27–68) Controls 62 range (40–74)		Atherosclerosis		MDA	Blood sample	HF: 9 (IQR 7.9–10.2) Controls: 7.7 (IQR 6.9–9.2)	<0.01
[44]	CHF 53 Controls 38		HFrEF. HFpEF	IHD, HVD, IDC, CCHD	Diuretics, digoxin and vasodilators.	MDA	Blood sample	HF: 10.3 (IQR 9–12) Controls: 7.9 (IQR 7–9)	<0.001
[45]	HF 29 (19 M/10 F) Controls 15 (8 M/7 F)		HFrEF	IHD		MDA	Blood sample	HF: 16 ± 7.48 Controls: 8 ± 3.16	<0.001
[46]	CHF 30 (13 M/17 F) Controls 55 (30 M/25 F)	CHF 73.1 ± 7.4, Controls 80 ± 17.4	HFrEF		Diuretics, BB, CAs, ACEi.	MDA	Blood sample	HF: 0.32 (0.21–0.52) Controls: 0.21 (0.17–0.25)	<0.001
[47]	HF 12 (9 M/3 F) Controls 25 (17 M/8 F)	HF 60.8 ± 4.6 Controls 56.0 ± 4.6	HFrEF	IHD		MDA	Blood sample	HF: 5.54 ± 2.29 Controls: 1.49 ± 0.85	<0.001
[48]	HF 29 (24 M/4 F) Controls 15 (10 M/5 F)	HF 61.9 ± 2.6	HFrEF	IHD, DCM	ACEi 38%, ARBs 38%, Diuretics 86%, Spironolactone 38%, Digoxin 66%, BB 24%	MDA	Blood sample	HF: 3.71 ± 0.10; Controls: 2.69 ± 0.12	<0.001
**Isoprostanes**									
[41]	HF 12 (9 M/3 F) Controls 25 (17 M/8 F)	HF 60.8 ± 4.6 Controls 56.0 ± 4.6	HFrEF	IHD		Isoprostanes	Urine sample	HF: 1930 ± 880 Controls: 350 ± 300	<0.001
[49]	HF 25 (22 M/3 F) Controls 25 (22 M/3 F)	HF 57 (27–75)	HFrEF	ICM, IDCM	ACEi 92%, LDs 88%, Digoxin 72%, Spironolactone 48%, BB 40%, ASA 40%, OAC 28%, Nitrates 4%	Isoprostaglandin F2α type III	Urine sample	HF: 2590 ± 1910 Controls: 1100 ± 360	<0.0001
[50]	CHF 39 (33 M/6 F) Controls 27 (21 M/6 F)	HF 66 ± 10 Controls 60 ± 10		IHD, IDCM, VHD	ACEi/ARBs 22%, BB 79%, ASA 38%, Diuretics 77%, Digoxin 46%, Spironolactone 54%, Warfarin 41%, Statins 67%	Isoprostanes	Blood sample	HF: 449 (IQR: 173–698) Controls: 82 (IQR: 53–95)	<0.001
[51]	CHF 30 (14 M/16 F) Controls 30 (18 M/12 F)		HFrEF, HFpEF	Ischaemic origin	Diuretics, BB, CCBs, ACEi	Isoprostanes 8,12-iso-iPF2α-VI	Blood sample	CHF: 356.1 ± 150.8 Controls: 136.3 ± 52.1	<0.0001
[52]	HF 20 (16 M/4 F) controls 20 (16 M/4 F)	HF 46 ± 7 controls 47 ± 6	HFrEF	DCM	Antiplatelets, ACE inhibitors, Nitrates, α-receptor blockers, CCBs, NSAIDs, Diuretics	Isoprostanes 8-epi-PGF2a	Urine sample	HF: 402 ± 40 Controls 236 ± 44	<0.001
[53]	HF 15 (7 M/8 F) Controls 15 (8 M/7 F)		HFrEF	HVD, HHD, IDCM, CHD	Nitrates n = 6, Ca antagonists n = 6, BB n = 3, ACEi n = 15, Digoxin n = 13, Diuretics n = 23	Isoprostane 15-F2t	urine sample	HF: 600 (IQR 355–720) Controls: 198 (IQR 125–281)	<0.001
[48]	HF 28 (24 M/4 F) controls 15 (10 M/5 F)	HF 61.9 ± 2.6	HFrEF	IHD, IDCM	ACEi 38%, ARBs 38%, Diuretics 86%, Spironolactone 38%, Digoxin 66%, BB 24%	isoprostanes 8-epi-PGF2a	Blood sample	HF: 234 ± 19 Controls: 133 ± 29	<0.01
**Myeloperoxidase**									
[54]	HF 102 (58%M/42%F) controls 105	HF 65 ± 14 Controls 44 ± 11	HFrEF	IHD		MPO	Blood sample	HF: 212 (160–262) Controls: 153 (138–178)	<0.0001
[55]	HF 46 (23 M/23 F) control 48 (24 M/24 F)	HF 72 ± 8 controls 73 ± 5	HFrEF			MPO	Blood sample	HF: 212 (160–262) Controls: 153 (138–178)	<0.05
[56]	HF 86 (42 M/44 F) controls 46	HF 73 (66–79)	HFrEF		ARBs 33%, ACEi 49%, Thiazide 16%, Potassium-sparing 21%, LDs 73%, CCBs 31%, BB 80%, Anticoags 55%, AP 34%, Statins 44%, Nitro 14%, Glucose-lowering 20%	MPO	Blood sample	HF: 101 (81–132) Controls: 86 (74–101)	<0.015
[57]	HF 55 (19 M/36 F) controls 18	HF 80 ± 8.7	HFrEF	ICM	LDs 76%, Mineralocorticoid receptor antagonists 33%, BB 62%, ACEi/ARBs 78%, Statins 64%	MPO	Blood sample	HF: 34.7 (22.7–44.0) Controls: 22.6 (18.2–32.0)	<0.026
[58]	HF 28 (22 M/6 F) Controls 1303 (730 M/573 F)	HF 68 (51–80) Controls 63 (45–80)	HFpEF	IHD	ACEi 88%, LDs 70%, Other diuretics 12%, BB 67%, Aldosterone antagonists 33%, CCBs 19%	MPO	Blood sample	HF: 49.5 (30.5–102.5) Controls: 27.3 (7.7–156.9)	<0.0001
[59]	HF 23 (20 M/3 F) Controls 14 (14 M)	HF 68 (51–80) Controls 63 (45–80)	HFrEF	ICM, IHD		MPO	Blood sample	HF: 33.6 (11.7–206.9) Controls: 18.3 (5.4–102.4)	<0.02
[60]	HF 285 (215 M/70 F) controls 35	HF 71.2 ± 11.3	HFrEF, HFpEF	IHD	ACEi/ARBs 82.8%, BB 69.8%, Digoxin 23.8%, ASA 67.7%, Spironolactone 54%, Diuretics 82.1%, Statins 62.4%	MPO	Blood sample	HF: 205.7 ± 272.6 Controls: 123± 170.5	=0.01
[61]	HF 68 (45 M/23 F) controls 10	HF 64.3 ± 13.4	HFrEF	IHD		MPO	Blood sample	HF: 9.3 ± 7 Controls: 4.19 ± 2	<0.007
[62]	HF 27 (14 M/13 F) Control 40 (29 M/11 F)	HF 64 (49–85) Control 66 (42–87)	HFrEF		Diuretics 51.85%, ACE 48.15, Cardiac glycosides 12.5%, Organic nitrate 4.17%, Statins 48.15%	MPO	Blood sample	HF: 1.1 (1.0–1.2) Controls: 0.80 (0.62–0.98)	*p* ˂ 0.05
**Nitrotirosine**									
[59]	HF 23 (20 M/3 F) Controls 14 (14 M)		HFrEF	ICM, IHD		Nitrotirosine	Blood sample	HF: 6.3 (0.0–67.6) Controls: 0.0 (0.0–45.0)	<0.02
[48]	HF 28 (24 M/4 F) 15 Controls (10 M/5 F)	HF 61.9 ± 2.6	HFrEF	IHD, IDCM	ACEi 38%, ARBS 38%, Diuretics 86%, Spironolactone 38%, Digoxin 66%, BB 24%	Nitrotirosine	Blood sample	HF: 469 ± 39 Controls: 313 ± 40	<0.05
[62]	HF 27 (14 M/13 F) Controls 50 (29 M/21 F)	HF 64 (49–85) Controls 66 (42–87)	HFrEF		Diuretics 51.85%, ACE 48.15, Cardiac glycosides 12.5%, Organic nitrate 4.17%, Statins 48.15%	Nitrotirosine	Blood sample	HF: 206.6 (154.6–307.2) Controls: 181.1 (114.4–234.4)	=0.0005
[63]	38 HF and 8 Controls	HF 55.5 ± 10.0 Controls 47.6 ± 13.0	HFrEF	IHD, DCM	ACEi 89.5%, digitalis 84.2%, diuretic 89.5%, and nitrates 23.6%	Nitrotirosine	Muscle tissue	HF: 13.5 ± 8.5 Controls: 2.0 ± 1.7	<0.001

HF: Heart Failure; CHF: Congestive heart failure, HFrEF: Heart Failure with Reduced Ejection Fraction; HFpEF: Heart Failure with Preserved Ejection Fraction; NYHA: New York Heart Association; IHD: Ischaemic Heart Disease; ICM: Ischaemic Cardiomyopathy; DCM: Dilated Cardiomyopathy; IDCM: Idiopathic Dilated Cardiomyopathy; HHD: Hypertensive Heart Disease; HVD: Heart Valve Disease; CHD: Congenital Heart Disease; ICHF: Ischaemic Congestive Heart Failure; NICHF: Non-Ischaemic Congestive Heart Failure; MI: Myocardial Infarction; ACEi: Angiotensin-Converting Enzyme Inhibitors; ARBs: Angiotensin Receptor Blockers; BB: Beta-Blockers; CCBs: Calcium Channel Blockers; MRAs: Mineralocorticoid Receptor Antagonists; AAs: Aldosterone Antagonists; OACs: Oral Anticoagulants; ASA: Acetylsalicylic Acid (Aspirin); AP: Antiplatelets; NSAIDs: Non-Steroidal Anti-Inflammatory Drugs; 8-OHdG: 8-Hydroxy-2′-deoxyguanosine, MDA: Malondialdehyde; MPO: Myeloperoxidase.

#### 3.2.2. 8-Hydroxy-2′-Deoxyguanosine

The forest plot reported in Figure 2 summarizes the results of studies comparing 8-OHdG levels in HF patients with healthy controls. Seven studies were included, encompassing 625 HF patients and 182 healthy controls. The meta-analysis revealed an overall RoM of 2.24 (95% CI: 1.75–2.88, *p* < 0.0001). Substantial heterogeneity was observed across the included studies, with an I^2^ value of 93.6%, reflecting significant variability in the results, though all of them showed higher levels of oxidative stress in HF patients. This variability may be attributed to differences in study populations, methodologies or biomarker measurement techniques. The visual analysis of funnel plot did not suggest asymmetry, a finding further supported by Begg’s test (*p* = 0.652). Sensitivity analysis did not find any influential studies.

#### 3.2.3. Telomere Length

The forest plot reported in Figure 3 compares telomere length in HF patients and healthy controls evaluated in 4 studies, comprising 721 HF patients and 293 healthy controls. The meta-analysis showed an overall RoM of 0.66 (95% CI: 0.53–0.81, *p* < 0.0001), indicating that HF patients have 34% shorter telomeres as compared to controls. Also, for this biomarker high heterogeneity was observed among the studies, with an I^2^ value of 94%. No evidence of asymmetry in the funnel plot was found. The sensitivity analysis indicated no influential studies

#### 3.2.4. Malondialdehyde

The forest plot shown in Figure 4 summarizes the results of a meta-analysis comparing MDA levels in HF patients and healthy controls. Twelve studies were included, involving 583 HF patients and 335 healthy controls. The pooled RoM was 1.87 (95% CI: 1.49–2.36, *p* < 0.0001), indicating that HF patients had, on average, an 87% higher mean MDA level compared to controls, reinforcing the evidence of elevated levels of oxidative stress in HF pathophysiology. Although all studies found significantly higher levels of MDA in HF patients, considerable heterogeneity was observed among studies, with an I^2^ value of 98.7%. Visual inspection of the funnel plot suggested a slight asymmetry, with a predominance of small studies reporting larger effect sizes. However, Begg’s test did not detect statistically significant asymmetry (*p* = 0.337), indicating no evidence of publication bias. Nevertheless, caution is warranted given the visual pattern and the limitations of Begg’s test in meta-analyses with a limited number of studies. No influential studies were found by the sensitivity analysis.

#### 3.2.5. Isoprostane

The forest plot summarizes the results of a meta-analysis aimed at comparing isoprostane levels in HF patients and healthy controls (Figure 5). Seven studies were included, involving 229 HF patients and 172 healthy controls. A pooled RoM of 2.83 (95% CI: 1.97–4.05, *p* < 0.0001) indicated that HF patients had, on average, a 183% higher mean isoprostane level compared to controls. Despite all studies showing significantly higher levels of isoprostane in HF patients, a high heterogeneity was observed (I^2^ = 94.2%). Visual inspection of the funnel plot suggested a slight asymmetry, with smaller studies tending to report larger positive effects. However, Begg’s test yielded a borderline *p*-value (*p* = 0.099), indicating that there was no statistically significant evidence of publication bias. Taken together, both the qualitative and quantitative assessments suggest at most a possible small-study effect, but not a confirmed asymmetry.

#### 3.2.6. Nitrotyrosine

The result of the comparison between nitrotyrosine levels in heart failure (HF) patients and healthy controls is shown in Figure 6. A total of five studies were included, involving 133 HF patients and 93 controls. The pooled RoM showed a strong association between elevated nitrotyrosine levels and oxidative stress in HF pathophysiology (RoM 2.19; 95% CI: 1.17–4.07, *p* = 0.0136). The heterogeneity among the studies (I^2^ = 86.2%), suggested considerable variability, although all of them showed a higher level of nitrotyrosine in the HF patients. No evidence of asymmetry or influential studies was revealed by correspondent analyses. The funnel plot appeared visually symmetrical, with a balanced distribution of studies around the estimated effect size. Begg’s test returned a non-significant *p*-value (*p* = 0.142), providing no statistical evidence of publication bias.

#### 3.2.7. Myeloperoxydase

The forest plot illustrates the comparative evaluation of myeloperoxidase (MPO) levels between HF patients and control subject (Figure 7). The number of studies included in the meta-analysis (nine) allowed to stratify HF patients into two subgroups: Group A patients with HFrEF and Group B patients with HFpEF. Group A comprises six studies involving a total of 607 patients and 294 controls. The forest plot shows that MPO levels are significantly higher in HFrEF patients compared to controls, with an overall RoM of 1.64 [95% CI: 1.27–2.11], and a high level of heterogeneity (I^2^ = 90%, *p* < 0.01). Group B includes three studies with 117 patients with HFpEF and 1335 controls. The RoM is not statistically significant, i.e., 1.23 [95% CI: 0.88–1.72], and the heterogeneity is moderate (I^2^ = 51%, *p* = 0.1283). The combined analysis across both groups includes a total of 724 HF patients and 1629 controls. The overall RoM is 1.52 [95% CI: 1.23–1.88], indicating significantly higher MPO levels in HF patients compared to controls. The high heterogeneity (I^2^ = 85%, *p* < 0.0001) underscores the variability between studies. The comparison between the two subgroups (HFrEF vs. HFpEF) did not reveal a statistically significant difference (χ^2^ = 1.03, *p* = 0.3110), although the RoM is slightly higher in the HFrEF subgroup (1.56 vs. 1.23). The smaller sample size and the moderate heterogeneity in this subgroup may account for the lack of statistical significance. The funnel plot, based on the overall meta-analysis without stratification, showed a mild asymmetry, with one outlier and a slightly unbalanced distribution of smaller studies (Figure 7). However, Begg’s test did not indicate statistically significant asymmetry (*p* = 0.144), suggesting limited evidence of publication bias. Given the relatively small number of studies and the high heterogeneity (I^2^ = 85%), the result should nonetheless be interpreted with caution.

## 4. Discussion

The field of biomarkers in HF is rapidly evolving, with numerous candidate markers recently highlighted by the Biomarker Study Group of the Heart Failure Association (HFA) of the European Society of Cardiology (ESC) [65]. In this systematic review and meta-analysis, we provide an updated synthesis of the literature on oxidative stress and DNA damage biomarkers, evaluating their potential roles in HF diagnosis, prognosis, and therapeutic targeting. Given the recent comprehensive analyses conducted by the HFA group, natriuretic peptides were excluded from this evaluation [66].

We assessed several oxidative stress and DNA damage biomarkers, including 8-hydroxy-2′-deoxyguanosine (8-OHdG), telomere length, malondialdehyde (MDA), isoprostanes, nitrotyrosine, myeloperoxidase (MPO), and DNA damage measured via the Comet assay and micronucleus analysis. Their consistent elevation in HF patients compared to healthy controls underscores their pathophysiological relevance and potential utility as indicators of oxidative injury.

Several key biomarkers—such as 8-OHdG, MDA, isoprostanes, and telomere shortening—have been linked to disease severity and all-cause mortality [58]. While natriuretic peptides and troponins remain central to HF diagnosis, oxidative stress biomarkers may provide additional insight into disease progression, risk stratification, and long-term monitoring.

Among the evaluated biomarkers, isoprostanes showed the largest increase (RoM = 2.83; 95% CI: 1.97–4.05), highlighting their robustness as indicators of lipid peroxidation. Similarly, MDA was significantly elevated (RoM = 1.87; 95% CI: 1.49–2.36), reinforcing the contribution of lipid oxidative injury in HF. DNA damage, as reflected by 8-OHdG (RoM = 2.24; 95% CI: 1.75–2.88), indicates genomic instability driven by ROS. Telomere shortening (RoM = 0.66; 95% CI: 0.53–0.81) further supports the association between chronic oxidative injury and cellular senescence. MPO (RoM = 1.46; 95% CI: 1.15–1.86), and nitrotyrosine (RoM = 2.19; 95% CI: 1.17–4.07) were also elevated, albeit to a lesser extent, suggesting a contributory role of inflammation and protein oxidation.

Figure 8 summarizes the results of the meta-analysis, based on 3015 HF patients and 2704 healthy controls, illustrating the RoMs for the biomarkers investigated. This visual summary highlights the relevance of oxidative stress and DNA damage in HF pathophysiology and supports their potential as prognostic tools or therapeutic targets.

Translating these findings from research into clinical practice remains a major challenge in elucidating the role of oxidative damage in the pathogenesis of heart failure (HF). Most of the biomarkers investigated exhibit a reliability of measurement (RoM) that is insufficient for their individual use in diagnosing or treating the disease. Nonetheless, the consistent associations observed across biomarkers reflecting different stages of the oxidative process provide compelling evidence for its involvement in both the onset and progression of HF.

The validation of early-effect biomarkers for clinical application encompasses methodological and epidemiological challenges. Key methodological considerations include the adoption of standardized measurement units and the establishment of universally accepted reference thresholds. From an epidemiological standpoint, emphasis should be placed on prospective studies and randomized controlled trials capable of unequivocally linking biomarker levels to disease risk. A comprehensive discussion of the validation process for early-effect biomarkers is provided by Bonassi and Fenech (2021) [67].

Future studies should investigate dynamic changes in oxidative stress biomarkers across different HF stages and in response to therapy, with the aim of identifying those most strongly associated with disease progression and clinical outcomes. While lipid peroxidation has been extensively studied, mechanisms such as oxidative DNA damage and peroxynitrite-mediated injury remain insufficiently explored and should be prioritized in future research. Standardization of biomarker assays and large-scale longitudinal studies are essential to establish their clinical utility in HF management.

### Study Limitations

This meta-analysis has several limitations that should be acknowledged. A major limitation of this meta-analysis is the high heterogeneity observed across studies (I^2^ > 90% for several biomarkers), likely attributable to differences in patient populations, biological matrices (blood vs. urine), and analytical methodologies. Despite subgroup and sensitivity analyses, residual heterogeneity remains, underscoring the need for standardized protocols in future research. Notably, only one out of the 44 studies evaluated failed to show increased biomarkers levels in HF patients, indicating that heterogeneity is driven primarily by differences in the magnitude of association (RoM ranged from 1.14 to 8.50), rather than by its direction. Another important limitation of the meta-analysis is the absence of stratified analyses by heart failure (HF) phenotype for most biomarkers. Considering the central role of HF phenotype in both diagnosis and prognosis, it is plausible that the strength of the association with oxidative damage may differ according to the degree of left ventricular ejection fraction. Furthermore, the predominance of cross-sectional studies substantially limits the ability to infer a causal relationship between oxidative damage and the pathogenesis of HF.

Some biomarkers, such as DNA damage assessed by the Comet assay or micronucleus formation, were reported in only a limited number of studies, precluding the calculation of pooled estimates. Although genomic instability is increasingly recognized as a component of HF pathophysiology, biomarkers showing this condition remain underrepresented in the literature. Expanded research in this area is critical to elucidate their prognostic value and therapeutic implications.

Overall, these findings underscore the importance of lipid peroxidation markers particularly isoprostanes and MDA which may potentially serve not only as diagnostic tools but also as therapeutic targets. In contrast, biomarkers associated with inflammation-mediated oxidative damage, such as MPO and nitrotyrosine, appear to play a secondary role, yet still warrant further investigation.

## 5. Conclusions

In summary, biomarkers of genomic instability show strong potential for improving the prediction and monitoring of heart failure progression, in line with the growing emphasis on personalized medicine. A major limitation, however, remains the lack of clinical validation of oxidative stress biomarkers. Their limited specificity currently prevents their use as standalone diagnostic tools. Nonetheless, when interpreted within specific clinical phenotypes, these biomarkers can provide valuable insights into the role of oxidative stress in heart failure pathophysiology. Stratifying patients accordingly may facilitate tailored therapeutic strategies and enhance prognostic accuracy. As highlighted by [61], the critical step for clinical validation is the implementation of prospective studies, randomized controlled trials, and pragmatic trials—an urgent objective that has yet to be achieved.

Rehabilitation is one field that could greatly benefit from this approach [1,67]. By demonstrating the consistent association between oxidative stress, DNA damage, and heart failure, this study highlights the potential of specific biomarkers to guide disease monitoring and therapeutic response during rehabilitation. In particular, lipid peroxidation and genomic instability markers emerge as promising prognostic tools, enabling patient stratification based on oxidative burden and supporting the design of personalized rehabilitation strategies. Markers such as MDA and MPO, which have demonstrated prognostic value and are implicated in oxidative stress–induced cardiomyopathy [55], may serve as actionable targets to evaluate and enhance the impact of interventions aimed at reducing oxidative stress and inflammation. Integrating these biomarkers into rehabilitation frameworks may therefore improve treatment effectiveness, refine risk prediction, and advance precision medicine in heart failure management.

This study is innovative as it provides the first comprehensive and systematic evaluation of oxidative stress and DNA damage biomarkers in heart failure through a rigorous meta-analysis. While previous research has focused mainly on natriuretic peptides, our work integrates a broad spectrum of biomarkers—including 8-OHdG, malondialdehyde, isoprostanes, telomere length, nitrotyrosine, and myeloperoxidase—thus offering a multidimensional characterization of oxidative injury at the genomic, lipid, and protein levels. By addressing these diverse biological pathways simultaneously, the study advances current knowledge of the role of oxidative stress in heart failure and highlights its prognostic and therapeutic potential.

The novelty of this work also lies in reconstructing the mechanistic link between oxidative stress, DNA damage, and cardiac dysfunction, moving beyond fragmented evidence from individual studies. The inclusion of genomic instability markers underscores a relatively unexplored but promising area with potential implications for risk stratification and personalized treatment. By tackling methodological heterogeneity and advocating for the standardization of biomarker assays, this study lays the groundwork for future validation efforts. Finally, its translational perspective—integrating oxidative stress biomarkers into rehabilitation frameworks—opens a new avenue for precision medicine in heart failure, broadening the clinical applicability of these findings.

## Figures and Tables

**Figure 1 antioxidants-14-01249-f001:**
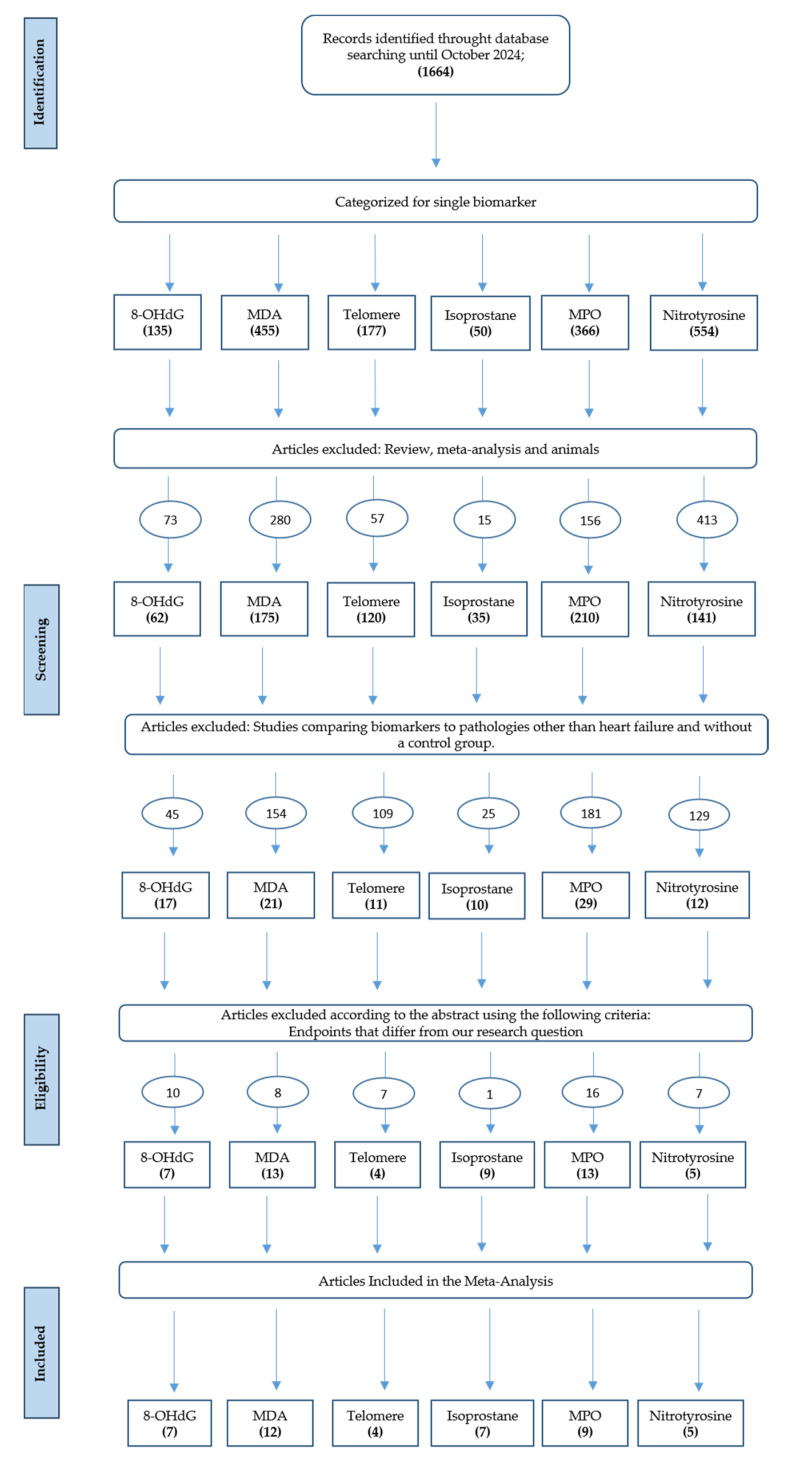
Flowchart of study selection for the meta-analysis.

**Figure 2 antioxidants-14-01249-f002:**
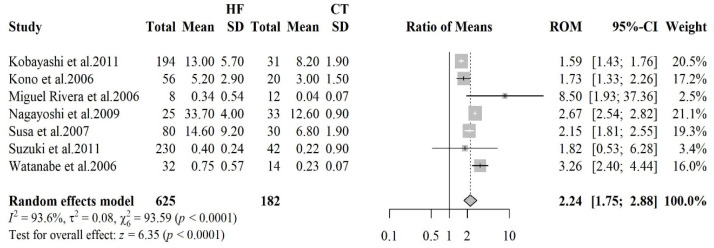
Forest plot summarizing studies on 8-hydroxy-2′-deoxyguanosine (8-OHdG) levels in heart failure patients versus healthy controls [26,27,28,29,30,31,32]. The results indicate significantly elevated oxidative DNA damage in HF patients.

**Figure 3 antioxidants-14-01249-f003:**
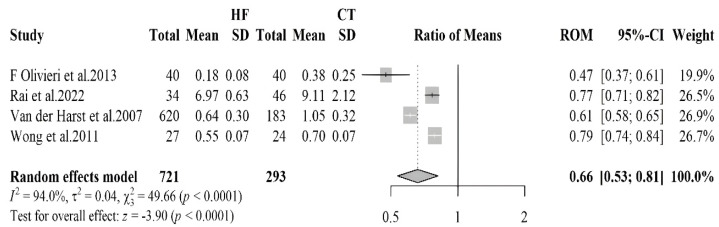
Forest plot summarizing studies on telomeres length in HF patients compared to healthy controls [33,34,35,36]. The analysis reveals significantly shorter telomeres in HF patients, suggesting increased cellular aging and oxidative stress.

**Figure 4 antioxidants-14-01249-f004:**
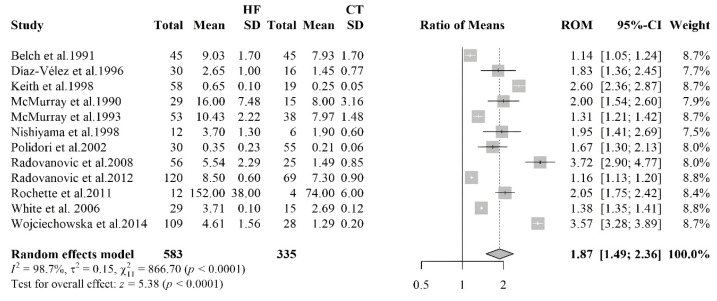
Forest plot summarizing studies on the levels of malondialdehyde (MDA) in HF patients and controls [37,38,39,40,41,42,43,44,45,46,47,48]. The results show higher MDA levels in HF patients, reflecting increased lipid peroxidation and oxidative stress.

**Figure 5 antioxidants-14-01249-f005:**
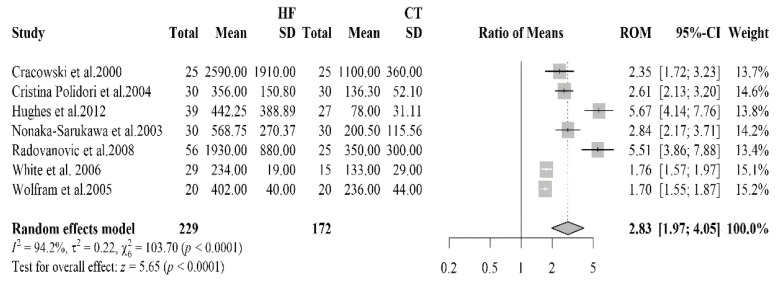
Forest plot summarizing studies on isoprostane levels in HF patients versus controls [47,48,49,50,51,52,53], indicating a nearly threefold increase in oxidative stress markers in HF patients.

**Figure 6 antioxidants-14-01249-f006:**
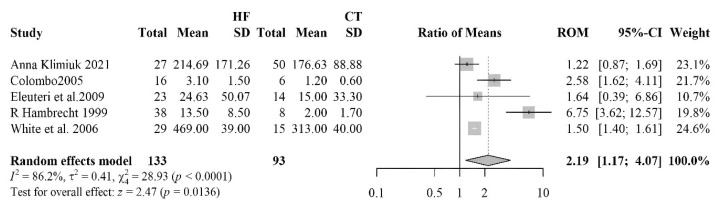
Forest plot summarizing studies on nitrotyrosine levels in HF patients and controls [48,59,62,63,64], demonstrating a significant increase in oxidative protein modifications in HF.

**Figure 7 antioxidants-14-01249-f007:**
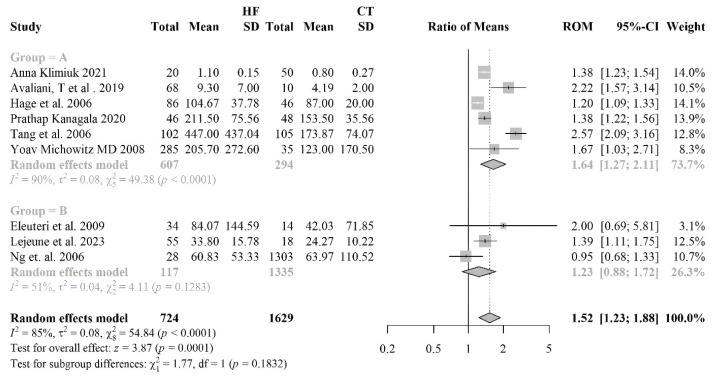
Forest plot comparing myeloperoxidase (MPO) levels between heart failure (HF) patients and healthy controls [54,55,56,57,58,59,60,61,62], stratified by HF phenotype. The analysis shows a significant increase in MPO levels, particularly among patients with HFrEF. Group A = HFpEF (heart failure with preserved ejection fraction); Group B = HFrEF (heart failure with reduced ejection fraction).

**Figure 8 antioxidants-14-01249-f008:**
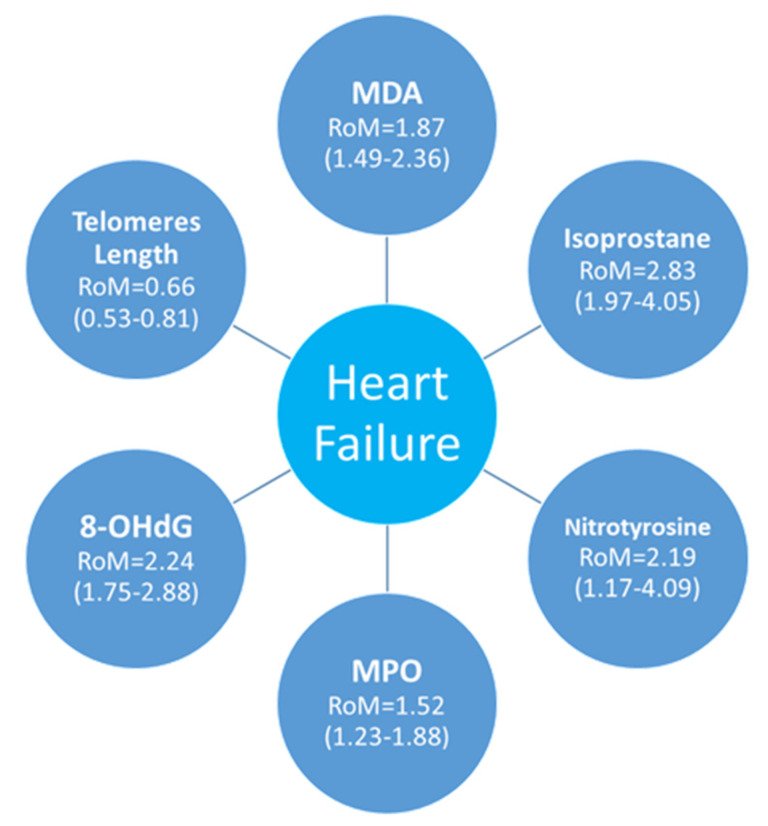
Overview of oxidative stress and DNA damage biomarkers in heart failure.

**Table 1 antioxidants-14-01249-t001:** Number of HF patients and controls analyzed per biomarker.

Biomarker	Studies Included (n)	HF Patients (n)	Controls (n)
**8-OHdG**	7	625	182
**MDA**	12	583	335
**Telomere**	4	721	293
**Isoprostanes**	7	229	172
**MPO**	9	724	1629
**Nitrotyrosine**	5	133	93
**Total**	44	3015	2704

## Data Availability

https://doi.org/10.5281/zenodo.17339301.

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
