# Peer review of "Oxidative Stress and DNA Damage Biomarkers in Heart Failure: A Systematic Review and Meta-Analysis"

_antioxidants, 2025, doi:10.3390/antiox14101249_

Round 1
Reviewer 1 Report
-
The introduction does not provide an overview of previously published systematic reviews or meta-analyses on similar topics. Please include a short paragraph situating the present study within the context of prior reviews, highlighting what is novel or different in this work.
-
There is a clear inconsistency in the reported sample sizes. The abstract states that 2,188 HF patients and 1,073 controls were included, whereas the discussion and figure summaries mention 3,015 HF patients and 2,704 controls. Please verify the correct numbers, clarify whether duplicate datasets were removed, and clearly explain the final counting rules in the Methods section.
-
The description of publication bias assessment is inconsistent. The abstract reports the use of Egger’s test, while the Methods section describes Begg’s test. Please unify the terminology and clearly state which test (or both) was applied.
-
The layout of Table 1 appears cluttered and difficult to follow. Please revise the formatting to improve readability, ensuring alignment of columns and consistency in the display of demographic and clinical variables.
-
The interpretation of funnel plots appears contradictory. For example, in the section on isoprostanes, the text first states that there is “no evidence of asymmetry” but later describes “moderate asymmetry” with a borderline p-value (p = 0.099). Please revise this section to provide a consistent interpretation, ensuring that visual inspection and statistical tests (Egger/Begg) are presented in a coherent manner.
-
In the “Data Availability” section, it is recommended to provide explicit information on how the underlying dataset and analytical scripts can be accessed (e.g., through an OSF link or as supplementary files). This will enhance transparency and reproducibility for readers and reviewers.
Refer to the above comments.
Reviewer 2 Report
This is a relevant contribution, which delves into the association between various biomarkers of oxidative stress, DNA damage and heart failure, and the need to use them in monitoring and prognosing the disease. The review is well written. I suggest changing the vertical format of Table 1 to horizontal. Since it probably won't fit on one page, it could be split into Table 1 for 8-hydroxy-2'-deoxyguanosine, Table 2 for Telomeres, Table 3 for malondialdehyde, Table 4 for isoprostanes, Table 5 for myeloperoxidase, and Table 6 for nitrotyrosine, or something similar.Author Response
Please see the attached document

Reviewer 3 Report
- There are obvious inconsistencies between the abstract and the main text regarding the number of included cases. For example, the abstract mentions data collected from 2,188 HF patients and 1,073 controls, but only 46 cases were finally included in the biomarker analysis after screening from 1,664 subjectsaccording to Figure 1. And these key numbers (2,188 and 1,073) even do not appear in the main text.
- The final number of included cases differs greatly from that reported in the abstract, which affects the credibility of the conclusions. If certain studies are ultimately not included in the analysis, they should be removed at the initial stage of literature collection.
- Different numbers of cases were included for different biomarkers. Readers have to calculate these numbers from the figures and tables themselves, which increases difficulty in interpretation. It is recommended to add a table clearly listing the actual number of cases included for each biomarker and to ensure consistency between the abstract and main text.
- Inconsistency between the abstract and the main text in logical order. For example, the analysis of isoprostanesappears in Figure 5 in the main text, but it is introduced first in the abstract, which affects reading logic.
- The Data Collection section is too general. It is recommended to provide a detailed description of the literature screening process and the determination of final sample sizes, specifying the number of cases for each analysis.
- In the main text, it ismentioned that“biomarkers of oxidative stress in HF patients: in total, the meta-analysis included 3,015 HF patients and 2,704 controls.” The relationship of these numbers to other numbers (e.g., 2,188, 1,664) should be explained to avoid reader confusion.
- In the legend of Figure 7, Group A and Group B should be explicitly defined as corresponding to HFpEF and HFrEF, respectively, for clarity.
Round 2
Reviewer 1 Report
The reviewer has no more concerns.
N/A